# Regulatory T cells in erythema nodosum leprosum maintain anti-inflammatory function

**Edessa Negera**[1,2]*, **Kidist Bobosha**[2], **Abraham Aseffa**[3], **Hazel M. Dockrell**[1], **Diana N. J. Lockwood**[1], **Stephen L. Walker**[1]

**1** London School of Hygiene and Tropical Medicine, Department of Clinical Research, London, United Kingdom, **2** Armauer Hansen Research Institute, Addis Ababa, Ethiopia, **3** World Health Organization, TDR, the Special Programme for Research and Training in Tropical Diseases, Geneva, Switzerland

* edessan@yahoo.com

**Data Availability Statement:** All relevant data are within the manuscript and its Supporting Information files.

## Abstract

### Background

The numbers of circulating regulatory T cells (Tregs) are increased in lepromatous leprosy (LL) but reduced in erythema nodosum leprosum (ENL), the inflammatory complication of LL. It is unclear whether the suppressive function of Tregs is intact in both these conditions.

### Methods

A longitudinal study recruited participants at ALERT Hospital, Ethiopia. Peripheral blood samples were obtained before and after 24 weeks of prednisolone treatment for ENL and multidrug therapy (MDT) for participants with LL. We evaluated the suppressive function of Tregs in the peripheral blood mononuclear cells (PBMCs) of participants with LL and ENL by analysis of TNFα, IFNγ and IL-10 responses to *Mycobacterium leprae (M. leprae)* stimulation before and after depletion of CD25$^+$ cells.

### Results

30 LL participants with ENL and 30 LL participants without ENL were recruited. The depletion of CD25$^+$ cells from PBMCs was associated with enhanced TNFα and IFNγ responses to *M. leprae* stimulation before and after 24 weeks treatment of LL with MDT and of ENL with prednisolone. The addition of autologous CD25$^+$ cells to CD25$^+$ depleted PBMCs abolished these responses. In both non-reactional LL and ENL groups mitogen (PHA)-induced TNFα and IFNγ responses were not affected by depletion of CD25$^+$ cells either before or after treatment. Depleting CD25$^+$ cells did not affect the IL-10 response to *M. leprae* before and after 24 weeks of MDT in participants with LL. However, depletion of CD25$^+$ cells was associated with an enhanced IL-10 response on stimulation with *M. leprae* in untreated participants with ENL and reduced IL-10 responses in treated individuals with ENL. The enhanced IL-10 in untreated ENL and the reduced IL-10 response in prednisolone treated individuals with ENL was abolished by addition of autologous CD25$^+$ cells.

**Funding:** This study is funded by Hospital and Homes of St Giles (UK) and Royal society of Tropical Medicine and Hygiene (RSTMH) small grant program (UK) to EN. The funders had no role in study design, data collection and analysis, decision to publish, or preparation of the manuscript.

**Competing interests:** The authors have declared that no competing interests exist.

## Conclusion

The findings support the hypothesis that the impaired cell-mediated immune response in individuals with LL is *M. leprae* antigen specific and the unresponsiveness can be reversed by depleting CD25$^+$ cells. Our results suggest that the suppressive function of Tregs in ENL is intact despite ENL being associated with reduced numbers of Tregs. The lack of difference in IL-10 response in control PBMCs and CD25$^+$ depleted PBMCs in individuals with LL and the increased IL-10 response following the depletion of CD25$^+$ cells in individuals with untreated ENL suggest that the mechanism of immune regulation by Tregs in leprosy appears independent of IL-10 or that other cells may be responsible for IL-10 production in leprosy. The present findings highlight mechanisms of T cell regulation in LL and ENL and provide insights into the control of peripheral immune tolerance, identifying Tregs as a potential therapeutic target.

## Author summary

Leprosy is complicated by episodes of inflammation called leprosy reactions. Leprosy reactions are important causes of nerve damage and illness. Erythema Nodosum Leprosum (ENL) also called type 2 reaction is a severe systemic immune-mediated complication of borderline and lepromatous leprosy. ENL causes high morbidity and thus requires immediate medical attention. We recruited 60 untreated patients with lepromatous leprosy (30 patients with ENL reactions and 30 patients without ENL reactions) in Ethiopia to better understand the loss of immune regulation in ENL. We took blood samples at 2 time points before and after prednisolone treatment and assessed if the regulatory T-cells in these patients are functionally competent to control inflammation. Previously we described that the proportion of Tregs are reduced in ENL. In the present study we found that despite the reduction in the proportion of Tregs, their functional integrity is intact and competent which confirms that ENL is associated with reduction of Tregs proportion but not with loss of their function. This is an important finding which suggests that future studies should focus on ways of increasing the proportion of Tregs in ENL to control the inflammation.

## Introduction

Leprosy is caused by *Mycobacterium leprae (M. leprae)*, an intracellular acid-fast bacillus. The disease mainly affects the skin, peripheral nerves, mucosa of the upper respiratory tract, and the eyes. The Ridley-Jopling (RJ) classification categorises leprosy into five forms which are determined by the host immune response. Tuberculoid (TT) leprosy and lepromatous leprosy (LL) are polar forms which are considered to be clinically stable [1]. TT leprosy is characterized by well-formed granulomas and a Th1 T cell response while LL is characterized by a lack of cell mediated immune responses [1,2].

Leprosy is complicated by episodes of inflammation called leprosy reactions. Leprosy reactions are a risk factor for nerve function impairment. These reactions are classified as Type I (reversal reaction; RR) or Type II (erythema nodosum leprosum; ENL) reactions. Type I reactions occur in all Ridley-Jopling types but particularly in borderline forms of the disease [3] whereas ENL affects individuals within borderline lepromatous leprosy (BL) and LL [4].

ENL is a multi-system disorder characterised by fever and painful skin nodules but neuritis, arthritis, iritis and other organ involvement may occur [5]. ENL has been considered to be due to immune complex deposition [6], although this is not consistently demonstrable. Histologically, neutrophils are the hallmark of ENL [7] but are not always present [8,9]. Neutrophil extracellular traps (NETs) formation in ENL skin lesions and increased selectin expression has been reported [10,11]. It remains unclear whether neutrophils initiate ENL or are recruited to ENL skin lesions.

There are several pieces of evidence for increased T cell activity in individuals with LL who have ENL compared to those without ENL. There is an increased of the CD4$^+$ to CD8$^+$ T cells ratio in ENL [12,13] and reduced numbers of regulatory T cells (Tregs) in individuals with ENL compared to those without ENL [13–15].

Tregs are important for the induction and maintenance of peripheral tolerance thereby preventing excessive immune responses and autoimmunity [16]. Tregs are a T cell subpopulation specialized for immune suppression and engaged in the maintenance of immunological self-tolerance and homeostasis [17]. Tregs induce suppressive immune-response through downregulating lymphocyte activation genes, inhibition of the proliferation and differentiation of T cells, upregulating regulatory cytokines such as TGFβ and IL-10 or by inducing apoptosis [18].

Tregs are defined as CD25 (interleukin-2 [IL-2] receptor α-chain)-expressing mainly CD4$^+$ T cells, which express the transcription factor forkhead box protein P3 (FoxP3), a master regulator for the development and function of Tregs [19] although there is no single marker universally used for Tregs. It has been reported that CD25 is expressed transiently on activated T cells which complicates the phenotypic identification of Tregs [20]

An exaggerated effector T-cell response with increased proinflammatory cytokines in ENL is associated with a reduced frequency of Tregs [13,21]. One study suggested that the downregulation of Tregs may favour the development of the T-helper-17 responses that characterize this reaction [22]. Analysis of the frequency of circulating Tregs using flow cytometry in PBMCs of six individuals with ENL showed that both the absolute count and percentage of Tregs were significantly lower in those with ENL (1.2%) compared to individuals with non-reactional LL (2.8%) [23]. Reduced numbers of Tregs and increased proinflammatory cytokine IL-17 production has also been reported in Brazilians with ENL [24]. Previous studies focused only on the number and proportion of Tregs in ENL and the association of the reduced number of Tregs with increased proinflammatory cytokines. However, none of these studies investigated the functional suppressive activity of Tregs in ENL.

The proportion of Tregs in non-reactional LL is higher compared to healthy individuals suggesting the lack of cell mediated immunity in LL may be mediated by Tregs. Previous studies demonstrated that macrophages are loaded with live *M. leprae* in untreated LL [25,26]. Infected macrophages with live *M. leprae* primed T cells towards the differentiation of Tregs and inhibition of Th1 cytokines and CD8$^+$ cytotoxicity [27,28]. A single-cell RNA sequencing has shown the expansion Tregs in the PBMCs of LL patients [29]. The histology of ENL skin lesion shows the majority of *M. leprae* in the macrophages are dead and fragmented [30]. The proportion of Tregs is lower in individuals with ENL compared to those with non-reactional LL and this reduction may play a role in the pathophysiology of ENL. However, it is unclear whether Tregs in non-reactional LL and ENL have normal functional activity. We recruited individuals with ENL and LL without ENL at ALERT Hospital, Ethiopia to evaluate the suppressive function of Tregs in peripheral blood of recruited participants.

## Materials and methods

### Ethical statement

Informed written consent for blood sample was obtained from participants following approval of the study by the London School of Hygiene & Tropical Medicine Research Ethics committee, UK, (#6391), Armauer Hansen Research Institute (AHRI) and All Africa Leprosy, Tuberculosis and Rehabilitation Training Centre (ALERT) Ethics review committee, Ethiopia (P032/12) and the National Research Ethics Review Committee, Ethiopia (#310/450/06). All participant data were analysed and reported anonymously.

### Study participants

Individuals with LL with or without ENL were recruited at ALERT Hospital, Addis Ababa, Ethiopia [9]. Leprosy was diagnosed and classified clinically and histologically using the Ridley–Jopling (RJ) classification. Children below 18 years old, adults above 65 years old, pregnant and lactating females, individuals with other clinical forms of leprosy (tuberculoid and borderline leprosy, Type 1 reaction) were excluded from the study.

### Blood sample collection, PBMCs isolation and storage

Twenty millilitres of venous blood were collected into sterile BD Heparinized Vacutainer tubes (BD, Franklin, Lakes, NJ, USA) and used for peripheral blood mononuclear cell (PBMC) isolation. Blood samples were obtained before prednisolone treatment of participants with ENL and after 24-weeks of prednisolone treatment. For LL participants without ENL reaction, blood samples were obtained before initiating WHO recommended MDT (monthly rifampicin, and daily Dapsone and Clofazimine) and after 24 weeks of MDT (after participants completed 50% MDT). Twenty-four weeks sampling in LL was chosen to match with the sampling time of individuals with ENL.

Peripheral blood mononuclear cells were separated by density gradient centrifugation at $800 \times g$ for 20 min on Ficoll–Hypaque (Histopaque, Sigma Aldrich, UK) as described earlier [13]. Cells were washed three times in sterile phosphate-buffered saline (Sigma Aldrich, UK) and re-suspended with 1mL of Roswell Park Memorial Institute (RPMI medium 1640 + GlutaMAX + Pen-Strep GBICO, Life technologies, UK). Cell viability was determined using 0.4% sterile Trypan Blue solution (Sigma Aldrich, UK); the viability was between 94 and 100%. PBMC freezing was performed using a cold freshly prepared freezing medium composed of 20% foetal bovine serum (FBS, heat inactivated, endotoxin tested $\leq 5$ EU/ml, GIBCO Life technologies, UK), 20% dimethyl sulfoxide in RPMI medium 1640. Cells were kept at $-80°C$ for 2 to 3 days and transferred to liquid nitrogen until use. Cell thawing was done as described by Thompson, Kunkel [31]. The procedure is briefly described as follows: cells were incubated in a water bath (37°C) until thawed halfway and re-suspended in 10% FBS in RPMI medium 1640 (1×) (37°C) containing 1/10,000 benzonase until completely thawed, washed two times (5 min each), and counted. The percentage viability obtained was above 90%.

### CD25+ cell separation

Frozen PBMC were thawed, washed and incubated with 20μl of CD25 micro beads II, human (Miteny Biotec, Bergisch Gladbach, Germany) in 80μl MACS buffer (Phosphate-buffered saline (PBS) with 0.5% Bovine serum albumin (BSA) and 2 mM EDTA) for 20 minutes at 4°C. Cells were washed and added to a MS column attached to a Magnetic Cell Sorter (MACS) (Milteny Biotec). CD25- cells were collected as flow through and the CD25+ population was collected by detaching the column from the magnetic cell sorter. Cells were washed with

MACS buffer and resuspended in AIM-V medium. The purity of the CD25⁻ and CD25⁺cell populations was >95% (S1 Fig).

### Lymphocyte stimulation tests (LST)

Total PBMCs (200,000 cells/well) were added in triplicate into 96-well U-bottom tissue culture plates and cultured with 10 mg/mL irradiated armadillo-derived *M. leprae* whole cell sonicate (was obtained through BEI Resources, NIAID, NIH: *Mycobacterium leprae*, Strain NHDP, Gamma-Irradiated Whole Cells (lyophilized), NR-19326), 1 mg/mL phytohemagglutinin (PHA) or AIM-V medium at 37˚C with 5% $CO_2$ and 70% humidity. After 6 days, supernatants were collected and kept frozen until used in enzyme-linked immunosorbent assay (ELISA).

Three groups of cells and two antigens (PHA and *M. leprae*) were used. In Group (i) intact PBMC (200,000 cells) were cultured in 200μl medium containing either PHA, *M. leprae* or AIM-V medium. In group (ii) CD25⁺ depleted PBMCs (200,000 cells) were cultured in 200μl medium containing either PHA, *M. leprae* or AIM-V and in group (iii) autologous CD25⁺ cells (25,000 cells) were added to CD25⁺ depleted PBMCs (175,000 cells) in 200μl medium containing either PHA, M. *leprae* or AIM-V. Cells from each group were cultured in triplicate.

### ELISA

Supernatants were analysed for cytokines using a Ready-Set-Go sandwich ELISA. Capture and biotinylated detection antibodies directed against IFN-γ, TNFα and IL-10, were purchased from Thermo Fisher Scientific, UK. A 96-well flat-bottom polystyrene MaxiSorp ELISA plates (Thermo Fisher scientific, UK) were used. Standards for each cytokine were prepared by serial dilution as recommended by the supplier. Detection was performed with avidin-horseradish peroxidase (Avidin-HRP) conjugated with tetramethylbenzidine following the supplier's procedure. For all plates, the optical density (OD) at 450 nm was measured using a SpectraMax plus microplate reader (Molecular device, UK). A curve fit was applied to each standard curve according to the manufacturer's manual. Sample concentrations were interpolated from these standard curves. The assay sensitivity was 4 pg/ml for IFN-γ and TNFα and 2 pg/ml for IL-10.

### Statistical analysis

Differences in cytokine concentrations were analysed with either unpaired/paired two-tailed t-tests or F-test using STATA v.17 (College Station, TX: Stata Corp LLC, USA). A Bonferroni correction was used for multiple comparisons. The statistical significance level was set at $p < 0.05$ and 95% confidence interval was constructed for each comparison.

## Results

Thirty participants with non-reactional LL and 30 LL participants with ENL were enrolled. The mean age of individuals with LL and ENL was 27 (sd = 9.7) and 26.2 (sd = 4.6) years respectively (P>0.05). Individuals with non-reactional LL or with ENL had comparable bacterial index (BI) at the time of leprosy diagnosis. Less than a quarter (23.3%) of non-reactional LL participants were females compared to half of those with ENL (P<0.05). Twenty-seven (90%) individuals with ENL had completed MDT at the time of recruitment. Seven (23.3%) individuals with uncomplicated LL had received MDT previously. Four (13.3%) had leprosy relapse and three (10.0%) had not completed a previous course of MDT (Table 1).

Table 1. Demographic characteristics of study participants.

| Category | LL (n = 30) | ENL (n = 30) | unpaired t-test |
|---|---|---|---|
| **Age (years)** | | | p = 0.5308 |
| Mean (sd) | 27.4(9.7) | 26.2 (4.60) | |
| Median (IQR) | 25(20–30) | 26 (22–29) | |
| **Sex n (%)** | | | p = 0.0323* |
| Male | 23(76.7) | 15(50.0) | |
| Female | 7(23.3) | 15(50.0) | |
| **BI at leprosy diagnosis** | | | p = 0.6644 |
| Mean (sd) | 4.3(1.1) | 4.13(1.1) | |
| Median (IQR) | 4.3(3.9–5.0) | 4.3 (3.15–5.15) | |
| **Previous MDT n (%)** | | | p <0.001* |
| Yes | 7(23.3) | 27(90.0) | |
| No | 22(76.7) | 3(10.0) | |
| **Weight (kg)** | | | P = 0.9154 |
| Mean(sd) | 53.5 (6.5) | 53.8 (11.8) | |
| Median (IQR) | 54.0 (47.5–58.5) | 54.0 (45.0–59.3) | |
| **ENL type n (%)** | | | |
| Acute | | 14(46.7) | |
| Recurrent | | 5(16.7) | |
| Chronic | | 11(36.7) | |
| **LL status at recruitment n (%)** | | | |
| Treated (relapse) | 4(13.3) | | |
| New | 23(76.7) | | |
| Discontinued | 3(10.0) | | |

sd = standard deviation; IQR = interquartile range; n = number; BI = Bacillary index; ENL = erythema nodosum leprosum; LL = lepromatous leprosy

*significant at 95% of confidence interval and α = 5%.

## Depletion of CD25$^+$ cells enhanced TNFα and IFNγ responses in LL participants

To analyse the role of CD25$^+$ cells in the production of proinflammatory cytokines (TNFα and IFNγ) and suppressor cytokine (IL 10), PBMCs from individuals with ENL (n = 30) and uncomplicated LL (n = 30) were depleted of CD25$^+$ cells. CD25$^+$ depleted PBMCs were stimulated with *M. leprae* whole cell sonicate (WCS) or PHA for 6 days cultured both with and without the reintroduction of autologous CD25$^+$ cells. Whole PBMC stimulation with *M. leprae* WCS or PHA were used as controls.

CD25$^+$ depleted PBMCs had an increased mean TNFα response (27.1 pg/ml) compared to control PBMCs (22.8pg/ml) before initiation of MDT (P = 0.02; 95% CI for mean difference = 0.5562 to 6.802) in individuals with uncomplicated LL. After 24 weeks of MDT, the TNFα response was significantly increased to 96.1pg/ml and 75.7pg/ml in CD25$^+$ depleted PBMCs and control PBMCs respectively (p = 0.0003; 95% CI = 9.634 to 31.220) (Fig 1). Similarly, the IFNγ response was higher in CD25$^+$ depleted PBMCs (96.5pg/ml) compared to control PBMCs (82.3pg/ml) in untreated LL participants (p = 0.008; 95% CI = 3.785–24.59). The IFNγ response increased tenfold, to 956.8pg/ml and 801.6pg/ml, in CD25$^+$ depleted PBMCs and control PBMCs respectively (p = 0.0078, 95% CI = 42.19–268.3) in samples taken after 24 weeks of MDT. In control PBMCs and in CD25$^+$ depleted PBMCs supplemented with autologous CD25$^+$ cells neither TNFα nor IFNγ production were significantly different after 24

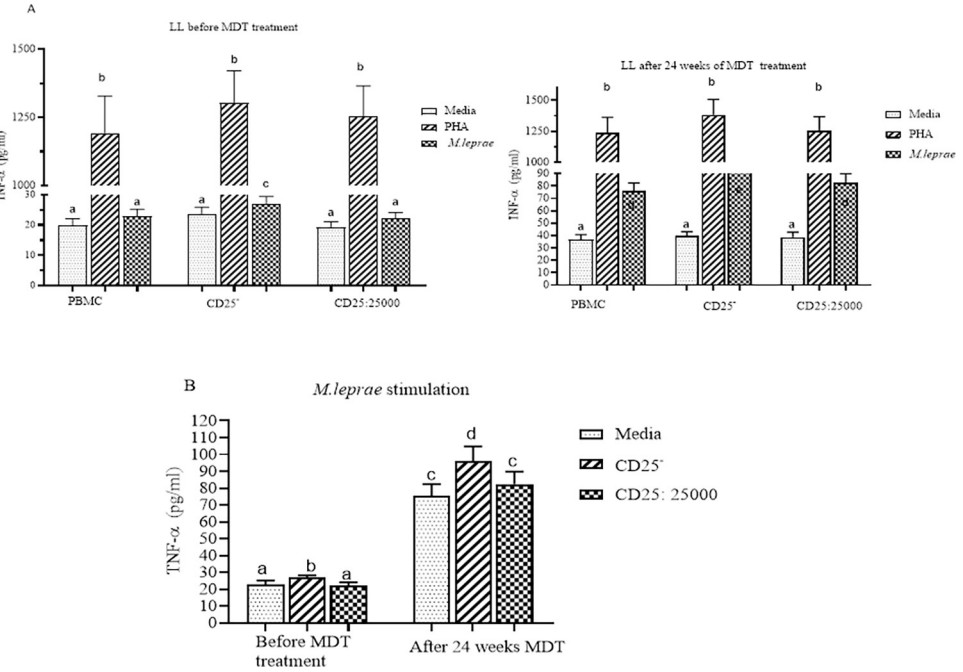

**Fig 1. The effect of depletion of CD25+ cells on TNFα secretion by PBMC from LL participants (n = 30).** (n = 30): (Fig 1A) TNFα response to PHA and *M. leprae* WCS stimulation before and after 24 weeks of MDT: (Fig 1B) TNFα response to *M. leprae* WCS stimulation in control PBMC, CD25+ depleted PBMC and with re-addition of CD25+ cells to CD25+ depleted PBMC from individuals with LL before and after 24 weeks of MDT. Each bar graph shows mean ± 95% CI. Any pair of different letters shows that the mean of TNFα production of the two groups are statistically significantly different at p = 0.05 and any pair of similar letters indicates that the mean of TNFα production the two groups are not statistically significantly different at P ≤0.05 and 95% confidence interval (CI). CI is calculated for mean difference. Media indicate AIM-V medium only and it was used in the assay as negative control and PHA used as a mitogen (positive control).

weeks of MDT compared to pre-treatment (Fig 2). The TNFα and IFNγ response to mitogen stimulation (PHA) was significantly increased in control and CD25+ depleted PBMCs after 24 weeks of MDT although the difference was not statistically significant (Figs 1 and 2).

## Depletion of CD25+ cells from PBMCs did not affect the IL-10 response in untreated individuals with LL

In individuals with LL, prior to starting MDT the IL-10 response to *M. Leprae* WCS stimulation in CD25+ depleted PBMCs was slightly higher (mean/median 217.2pg/ml) compared to that in control PBMCs (184.8pg/ml) but the difference was not statistically significant. After 24 weeks of MDT, the IL-10 response to *M. leprae* WCS stimulation was reduced in both CD25+ depleted PBMCs and control PBMCs but did not reach statistical significance. There was no evidence of a difference in IL-10 responses to *M leprae* WCS stimulation in CD25+ depleted PBMCs or control PBMCs after 24 weeks of MDT compared to pre-treatment in individuals with uncomplicated LL (Fig 3).

## Depletion of CD25+ cells enhanced TNFα and IFNγ responses in individuals with ENL

Depletion of CD25+ cells from PBMCs was associated with an enhanced TNFα response to *M. leprae* stimulation in individuals with ENL before and after 24 weeks of prednisolone

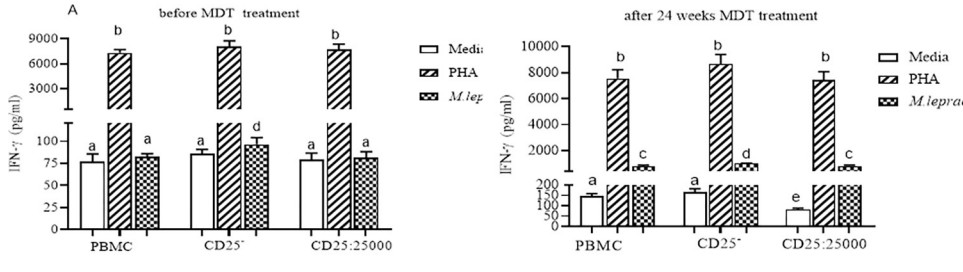

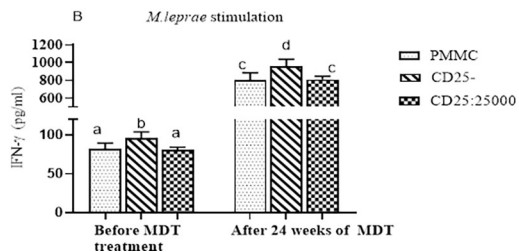

**Fig 2.** The effect of depletion of CD25[+] cells on IFNγ secretion by PBMC from LL participants (n = 30): (Fig 2A) IFNγ response to PHA and *M. leprae* WCS stimulation before and after 24 weeks of MDT: (Fig 2B) IFNγ response to *M. leprae* WCS stimulation in control PBMC, CD25+ depleted PBMC and autologous CD25[+] cells re-added to CD25[+] depleted cells from individuals with uncomplicated LL before and after 24 weeks of MDT. Each bar graphs shows mean ± 95% confidence interval (CI). Any pair of different letters shows that the mean of IFNγ production of the two groups are statistically significantly different at p = 0.05 and any pair of similar letters indicates that the mean of IFNγ production the two groups are not statistically significantly different at P ≤0.05 and 95% CI. CI is calculated for mean difference. Media indicate AIM-V medium only and it was used in the assay as a negative control and PHA used as a mitogen (positive control).

treatment. In CD25[+] depleted PBMCs TNFα production following *M. leprae* stimulation was significantly enhanced (1374 pg/ml) compared with whole PBMCs (1,008 pg/ml) before treatment with prednisolone (P<0.001; 95% CI = 163.0 to 570.7). Similarly, TNFα production was significantly increased (1581 pg/ml) in CD25[+] depleted PBMCs compared with control PBMCs (1,159 pg/ml) after completion of prednisolone treatment (P<0.001; 95% CI = 187.4 to 656.4). There was a trend towards increased mitogen (PHA) induced TNF production when CD25[+] cells were depleted from PBMC before and after prednisolone treatment (Fig 4).

IFNγ production was significantly enhanced in CD25[+] depleted PBMCs from ENL patients (10,183 pg/ml) compared to control PBMCs (7833 pg/ml) before prednisolone treatment (P<0.001; 95% CI = 1,164 to 3,536). After prednisolone treatment of individuals with ENL, IFNγ production was increased to 11,711 pg/ml in CD25[+] depleted PBMCs compared to 9,008 pg/ml in control PBMCs in response to *M. leprae* WCS stimulation (P <0.001; 95% CI = 1,338 to 4,067). CD25[+] cells depletion did not affect the mitogen induced IFNγ response in individuals with ENL (Fig 5).

## Differential IL-10 responses in CD25[+] depleted PBMCs before and after prednisolone treatment of ENL

The IL-10 response to *M. leprae* WCS stimulation was significantly enhanced in CD25[+] depleted PBMCs (199.9pg/ml) compared to control PBMCs (148.1pg/ml) in participants with ENL before prednisolone treatment (p = 0.001; 95% CI = 21.65 to 82.01). However, after completing prednisolone treatment, the IL-10 response to *M. leprae* WCS stimulation was significantly reduced in CD25[+] depleted PBMCs (127.7pg/ml) compared to control PBMCs

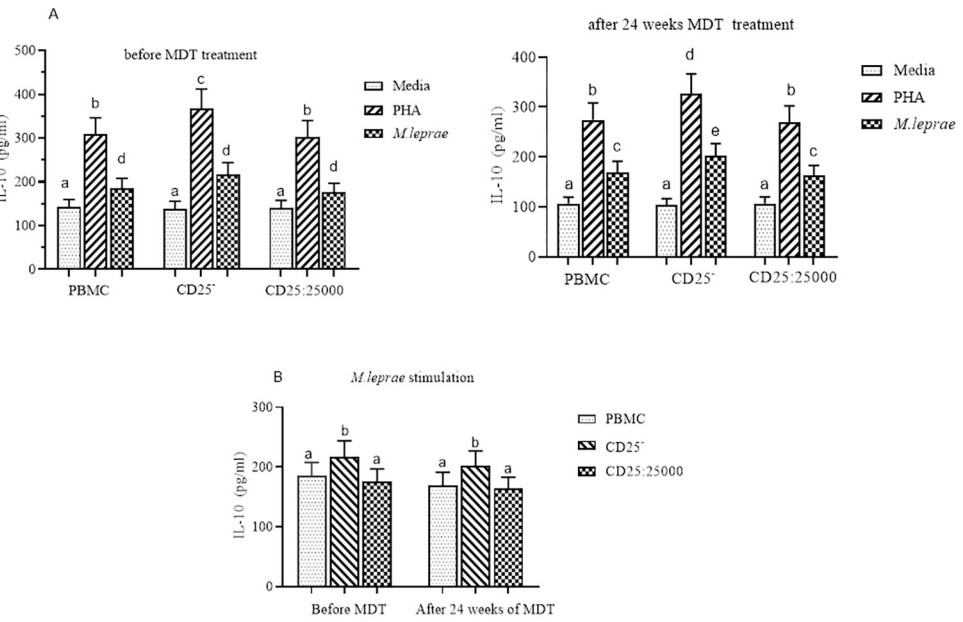

**Fig 3.** The effect of depletion of CD25+ cells on IL-10 secretion by PBMC from LL participants (n = 30): (Fig 3A) IL-10 response to PHA and *M. leprae* WCS stimulation before and after 24 weeks of MDT: (Fog 3B) IL-10 response to *M. leprae* WCS stimulation in intact PBMC, CD25+ depleted PBMC and CD25+ cells re-added to CD25+ depleted cells from uncomplicated LL before and after 24 weeks of MDT. Each bar graphs shows mean ± 95% Confidence interval (CI). CI is calculated for mean difference. Any pair of different letters shows that the mean of IL-10 production of the two groups are statistically significantly different at p = 0.05 and any pair of similar letters indicates that the mean of IL-10 production the two groups are not statistically significantly different at P ≤0.05 and 95% CI. Media indicate AIM-V medium only and it was used in the assay as negative control and PHA used as a mitogen (positive control).

(170.3pg/mL) (p = 0.0016;95% CI = -68.40 to -16.76). In control PBMCs and CD25+ depleted PBMCs supplemented with autologous CD25+ cells, the IL-10 response to *M. leprae* WCS stimulation was not statistically significantly different before and after completion of prednisolone treatment of ENL (Fig 6).

## Discussion

Tregs are essential for maintaining peripheral tolerance, preventing autoimmune disease and limiting chronic inflammatory diseases. They may also inhibit sterilizing and antitumor immunity which is not a desirable effect of Tregs [32].

We have shown that TNFα and IFNγ secretion was significantly enhanced in CD25+ depleted PBMCs in individuals with LL in response to the stimulation with *M. leprae* WCS both before and after 24 weeks of MDT. The TNFα response was found to be generally low in CD25+ depleted PBMCs in untreated individuals with LL compared with after 24 weeks of MDT. The increased TNFα and IFNγ response by the CD25+ depleted PBMCs to *M. leprae* WCS stimulation compared to control PBMCs in untreated LL suggests that Tregs may contribute to the lack of inflammatory response to *M. leprae* in LL by downregulating proinflammatory cytokines. One study has reported that the T cell response to *M. leprae* specific antigen stimulation was increased in CD25+ depleted PBMCs in 40% of untreated individuals with LL [33]. The IFNγ response was significantly increased in CD25+ depleted PBMCs while CD25+FoxP3+ Tregs were downregulated following stimulation with *M. leprae* specific antigen [33].

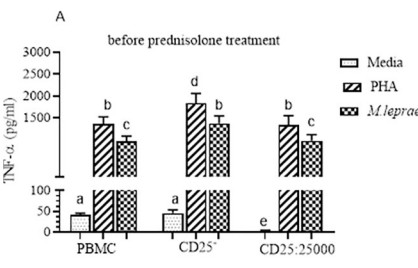
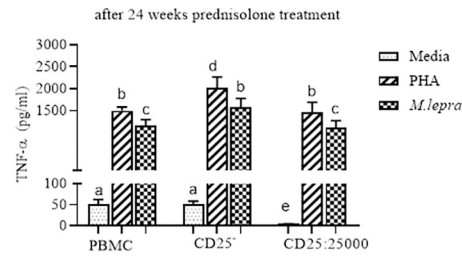

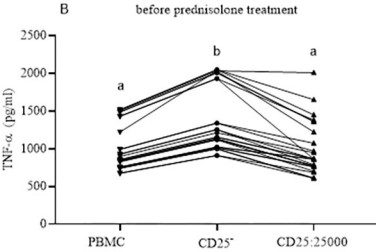
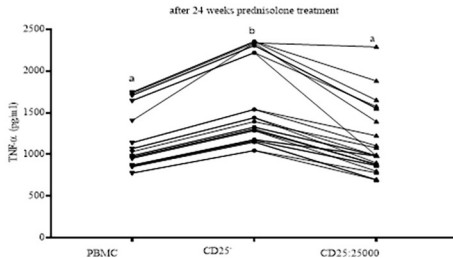

**Fig 4.** The effect of depletion of CD25+ cells on TNFα secretion by PBMC from ENL participants (n = 30): (Fig 4A) TNFα response to PHA and *M. leprae* WCS stimulation before and after 24 weeks prednisolone treatment of individuals with ENL; (Fig 4B) TNFα response to *M. leprae* WCS stimulation in control PBMC, CD25+ depleted PBMCs and CD25+ re-added cells to CD25+ depleted PBMC from individuals with ENL before and after 24 weeks of prednisolone treatment. Each bar and line graph show mean ± 95% confidence interval (CI). CI is calculated for mean difference. Any pair of different letters shows that the mean of TNFα production of the two groups are statistically significantly different at p = 0.05 and any pair of similar letters indicates that the mean of TNFα production the two groups are not statistically significantly different at P ≤0.05 and 95% CI. Media indicate AIM-V medium only and it was used in the assay as negative control and PHA used as a mitogen (positive control).

Reduced number or loss of suppressive function of Tregs have been reported in autoimmune diseases such as in diabetes mellitus [34], multiple sclerosis [35] and systemic lupus erythematosus [36]. In these autoimmune diseases, Tregs are unable to control the autoreactive immune response to self-antigens and the sustained immune response leads to chronic inflammatory damage. [37]. Increased numbers of Tregs are also associated with pathological processes. The percentage of Tregs are increased in lymph nodes associated with the site of *Mycobacterium tuberculosis* infection [38,39] with an associated delay in the activation of effector T cells in draining lymph nodes. [40].

Tregs downregulate effector T cells primarily to limit the host tissue damage caused by the immune responses against pathogens. However, this situation may lead to the prolonged survival of the pathogens in the host [41]. Increased numbers of Tregs are implicated in the hindering of effector T cell migration to the lungs in individuals with pulmonary tuberculosis [40]. Leprosy is a chronic bacterial infection. LL is characterized by uncontrolled *M. leprae* multiplication in the host cells with an associated lack of cell-mediated immune response [2]. Increased proportion of Tregs in LL are associated with T-cell unresponsiveness and persistence of *M. leprae* in the host [33]. We and others have previously shown that stimulation of PBMCs from untreated LL with *M. leprae* antigen does not induce proliferation of T-effector cells [13,33].

The significant increase of TNFα and IFNγ responses to *M. leprae* stimulation after 24 weeks of MDT in LL may be associated with the killing of *M. leprae*. The first dose of MDT reportedly kills at least 99% of the *M. leprae* [42]. Clofazimine and rifampicin are bactericidal

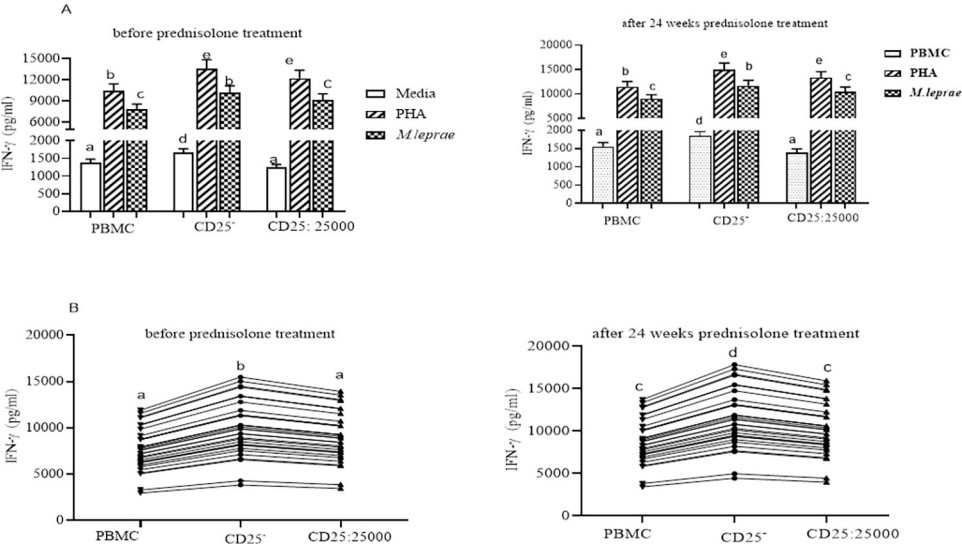

**Fig 5.** The effect of depletion of CD25+ cells on IFNγ secretion by PBMC from ENL participants (n = 30): (Fig 5A) IFNγ response to PHA and *M. leprae* WCS stimulation before and after 24 weeks of prednisolone treatment; (Fig 5B) IFNγ response to *M. leprae* WCS stimulation in control PBMC, CD25+ depleted PBMC and autologous CD25+ cells added PBMC to CD25+ depleted PBMC from individuals with ENL before and after 24 weeks of prednisolone treatment. Each bar and line graphs show mean ± 95% confidence interval (CI). Any pair of different letters shows that the mean of IFNγ production of the two groups are statistically significantly different at p = 0.05 and any pair of similar letters indicates that the mean of IFNγ production the two groups are not statistically significantly different at P ≤0.05 and 95% CI. Media indicates AIM-V medium only which was used in the assay as negative control with PHA used as a positive control.

antibiotics which kill *M. leprae*. Clofazimine also increases lipid metabolism in *M. leprae* infected cells thereby promoting the differentiation and proliferation of effector T-cells [43]. Clofazimine induces IFNγ production and downregulates suppressor T-cells in *M. leprae*-infected cells suggesting the effectiveness of clofazimine against leprosy is through modulation of lipid metabolism and activation of effector immune repose in *M. leprae*-infected host cells [44]. The interaction between MDT and Tregs needs to be further explored.

The significantly reduced TNFα and IFNγ response to *M. leprae* stimulation in the control (intact) PBMCs before MDT in LL shows the unresponsiveness of cell mediated immunity in these individuals and is consistent with previous findings of T cell anergy to *M. leprae* antigens in untreated LL [45,46]. The mechanism of this tolerance (anergy) and the role of Tregs needs further investigation.

In the present study, we have shown that MDT treatment and depletion of CD25+ cells are associated with the restoration of the T cell response to *M. leprae* stimulation. The killing of *M. leprae* by MDT and the depletion of regulatory T-cells may contribute to the apparent restoration of cell- mediated immune response to *M. leprae* antigen stimulation of PBMCs from individuals with LL. We and others have previously shown a reduced percentage of Tregs in ENL compared to participants with uncomplicated LL and endemic healthy controls [13–15,23]. However, whether there is concomitant functional impairment (loss of functional integrity of Tregs) has not been investigated.

TNFα and IFNγ secretion were significantly increased in CD25+ depleted PBMCs in individuals with ENL in response to *M. leprae* WCS stimulation before and after completion of 24 weeks prednisolone treatment. The TNFα response was higher in CD25+ depleted PBMCs compared to control PBMCs following stimulation with *M. leprae* WCS. The TNFα response was significantly decreased in autologous CD25+ supplemented PBMCs confirming that the

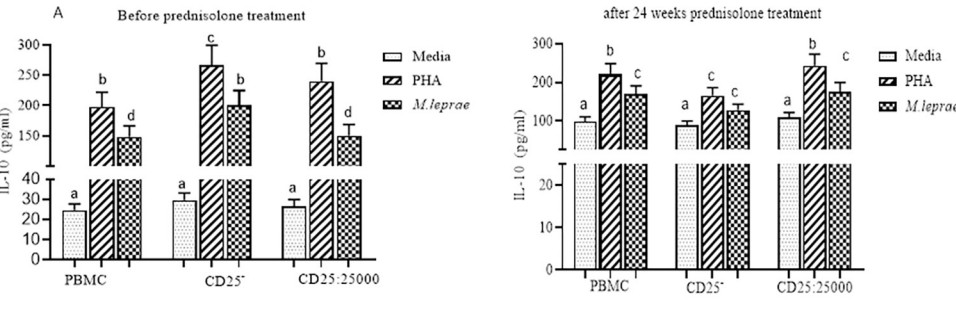

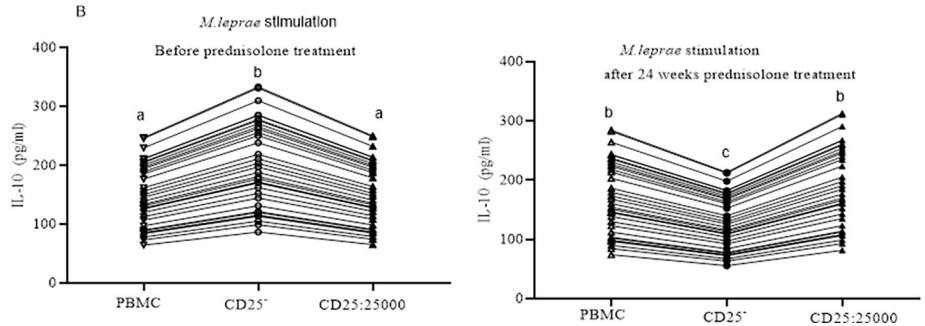

**Fig 6.** The effect of depletion of CD25[+] cells on IL-10 secretion by PBMC from ENL participants (n = 30): (Fig 6A) IL-10 response to PHA and *M. leprae* WCS stimulation before and after 24 weeks of prednisolone treatment of individuals with ENL; (Fig 6B) IL-10 response to *M. leprae* WCS stimulation in control PBMCs, CD25[+] depleted PBMCs and autologous CD25[+] re-added PBMCs from individuals with ENL before and after 24 weeks of prednisolone treatment. Each bar and line graphs show mean ± 95% confidence interval (CI). Any pair of different letters shows that the mean of IL-10 production of the two groups are statistically significantly different at p = 0.05 and any pair of similar letters indicates that the mean of IL-10 production the two groups are not statistically significantly different at P ≤0.05 and 95% CI. AIM-V medium only was used in the assay as the negative control and PHA used as positive control.

suppressive function of CD25[+] T cells is intact in ENL despite their reduction in number. Similarly, the IFNγ response was significantly enhanced in CD25[+] depleted PBMCs from untreated individuals with ENL compared to the control PBMCs and the response was reduced by the reintroduction of autologous CD25[+] cells to CD25[+] depleted PBMCs. A similar response was obtained in samples taken following completion of 24 weeks of prednisolone treatment in individuals with ENL.

Corticosteroid treatment is used to control inflammation. Prednisolone treatment of inflammatory disease is associated with increased circulating Tregs and the anti-inflammatory roles of prednisolone are thought to be mediated, in part, by Tregs via a miR-342-dependent mechanism [47]. Our group and others have described increased percentages of Tregs after prednisolone treatment of ENL [13,14]. In the present study, we report increased levels of pro-inflammatory cytokines (TNFα and IFNγ) in CD25[+] depleted PBMCs before and after prednisolone treatment and this was abolished by the re-addition of autologous CD25[+] cells to CD25[+] depleted PBMCs. This confirms that the relative numerical reduction of Tregs in ENL [13] is not associated with a dismissed ant-inflammatory function.

The ability of Tregs to suppress proinflammatory cytokines in leprosy may be achieved by inhibiting the production of T-cell derived cytokines, by inhibiting monocyte-driven proin-flammatory cytokines or through direct apoptosis of activated T cells. Thus, the mechanism

through which Tregs downregulate pro-inflammatory cytokines (TNFα and IFNγ) in *M. leprae WCS* stimulated cells in LL and ENL needs further investigation.

Together these findings suggest that Tregs in ENL are not functionally impaired and the development of ENL is associated with the reduced proportion of Tregs. It could be hypothesised that after *M. leprae* infects and multiplies in the host macrophages, it primes T cells towards Tregs differentiation. Previous studies have reported that *M. leprae* cell wall lipids induced immune suppression in mice [48]. Increased numbers of Tregs downregulate the production of Th1 cytokines. The inhibition of Th1 cytokine production reduces macrophage, T-effector cell and neutrophil killing of *M. leprae*. Treatment of LL with MDT kills *M. leprae* and the antigenic components from killed *M. leprae* induce the proliferation of Th1 T-cells which in turn restore Th1 cytokines and thereby reduce the proportion of Tregs. Antigens from *M. leprae* killed by MDT activates effector T-cells and in some individuals with LL in association with the reduction in the proportion of Tregs leads to the development of ENL. However, the underlying mechanism has not been fully elucidated and this does not explain why a small proportion of individuals develop ENL prior to MDT.

Treatment of ENL with prednisolone is associated with an increased proportion of Tregs which leads to the inhibition of inflammation [49]. It is suggested that prednisolone controls inflammation via TGFβ and TGFβ suppresses the functions of Th1 effector cells and promotes the generation of Tregs [50,51].

Interestingly, the anti-inflammatory IL-10 response to *M. leprae* WCS stimulation was not affected by the depletion of CD25[+] cells before or after the initiation of MDT in LL. This suggests that either the suppressive function of Tregs in LL is independent of IL-10 or that other sources such as Th1 and innate cells are responsible for IL-10 production [52]. In active pulmonary tuberculosis IFNγ and IL-10 are produced by Th1 T cells supporting the fact that IL-10 production is not confined to Tregs and Th2 cells [53]. Elucidating the sources of IL-10 in LL could shed light on the mechanism of T cell regulation in leprosy.

Unlike for the proinflammatory cytokines, a differential IL-10 response was observed in response to *M. leprae* WCS stimulation. The IL-10 response to *M. leprae* WCS stimulation was enhanced in CD25[+] depleted cells compared to that in control PBMCs of untreated individuals with ENL and was significantly reduced after prednisolone treatment. These results were opposite to what we had predicted. IL-10 is a regulatory cytokine and we expected that depleting CD25[+] cells would deplete IL-10 producing Tregs and hence the IL-10 response to *M. leprae* WCS stimulation is downregulated. The enhanced IL-10 response in CD25[+] depleted PBMCs to *M. leprae* WCS stimulation in untreated ENL could be explained by one of the following two possibilities: either the immune-regulation in untreated ENL is through an IL-10 independent mechanism such as suppression by cytolysis, suppression by metabolic disruption or suppression by modulation of dendritic-cell (DC) maturation and function- highlighting the need for further investigation of the mechanism of immune regulation in ENL; or the enhanced IL-10 in CD25[+] depleted PBMCs of untreated ENL is derived from non-Treg sources of IL-10 such as monocytes, macrophages, dendritic cells, neutrophils, mast cells or Th1 cells [54].

In conclusion, we confirmed that T cell unresponsiveness in LL is *M. leprae* antigen specific and can be reversed after treatment of LL with MDT or by depleting Tregs. The study has shown that Tregs cells do not affect the IL-10 response to *M. leprae* in individuals with LL. We have also shown that the suppressive function of Tregs in ENL is very likely intact. The pathogenesis of ENL is apparently associated with decreased number of Tregs but not with loss of function.

Our findings help to delineate the mechanisms of T cell regulation in different clinical forms of leprosy particularly in ENL. This may provide insights into the control processes of peripheral immune tolerance and yield potential therapeutic targets.

## Supporting information

**S1 Fig. Flowcytometry data dot plot analysis of bulk (total PBMCs), CD25 depleted and CD25 positive population.** Using magnetic sorter, fractions of each cell population were analysed for their expression of CD 3, CD4 and CD25.
(TIF)

## Acknowledgments

We would like to thank the study participants for donating blood samples and their time. We are grateful to BEI Resources, University Boulevard, Manassas, USA for providing Gamma-Irradiated *Mycobacterium leprae* whole cells.

We would like to thank the staff of the red medical clinic of ALERT hospital, Ethiopia in particular, the Sr. Genet Amare, Sr. Haregewoin, Mr. Fikre Mekuria and Mr. Yilma Tesfaye. This study would have not been possible without the administrative support of LSHTM and AHRI.

## Author Contributions

**Conceptualization:** Edessa Negera, Hazel M. Dockrell, Diana N. J. Lockwood, Stephen L. Walker.

**Data curation:** Edessa Negera, Kidist Bobosha, Abraham Aseffa, Hazel M. Dockrell, Diana N. J. Lockwood, Stephen L. Walker.

**Formal analysis:** Edessa Negera.

**Funding acquisition:** Edessa Negera.

**Investigation:** Edessa Negera, Kidist Bobosha, Stephen L. Walker.

**Methodology:** Edessa Negera, Kidist Bobosha, Abraham Aseffa, Hazel M. Dockrell, Diana N. J. Lockwood, Stephen L. Walker.

**Project administration:** Edessa Negera, Stephen L. Walker.

**Resources:** Edessa Negera, Kidist Bobosha.

**Software:** Edessa Negera.

**Supervision:** Abraham Aseffa, Hazel M. Dockrell, Diana N. J. Lockwood, Stephen L. Walker.

**Validation:** Edessa Negera.

**Visualization:** Edessa Negera.

**Writing – original draft:** Edessa Negera.

**Writing – review & editing:** Edessa Negera, Kidist Bobosha, Abraham Aseffa, Hazel M. Dockrell, Diana N. J. Lockwood, Stephen L. Walker.

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
