## [Decision Letter · Decision Letter 0]

9 May 2022

Dear Dr Negera,

Thank you very much for submitting your manuscript "Regulatory T cells in Erythema nodosum leprosum maintain anti-inflammatory function" for consideration at PLOS Neglected Tropical Diseases. As with all papers reviewed by the journal, your manuscript was reviewed by members of the editorial board and by several independent reviewers. In light of the reviews (below this email), we would like to invite the resubmission of a significantly-revised version that takes into account the reviewers' comments. 

The author should carefully addressed all the comments from reviewers as the results have not clearly and sufficient to cover the title of manuscript. More experiments and data need to be performed according to the comments.

We cannot make any decision about publication until we have seen the revised manuscript and your response to the reviewers' comments. Your revised manuscript is also likely to be sent to reviewers for further evaluation.

Sincerely,

Furen Zhang

Associate Editor

Gerson Penna

Deputy Editor

The author should carefully addressed all the comments from reviewers as the results have not clearly and sufficient to cover the title of manuscript. More experiments and data need to be performed according to the comments.

Reviewer's Responses to Questions

**Key Review Criteria Required for Acceptance?**

**Methods**

-Are the objectives of the study clearly articulated with a clear testable hypothesis stated?

-Is the study design appropriate to address the stated objectives?

-Is the population clearly described and appropriate for the hypothesis being tested?

-Is the sample size sufficient to ensure adequate power to address the hypothesis being tested?

-Were correct statistical analysis used to support conclusions?

-Are there concerns about ethical or regulatory requirements being met?

Reviewer #1: YEs

Reviewer #2: 1. Please add figure of flow cytometry dot plot showing CD25+ and CD25neg depletion assay.

2. Treg cells are major source of IL-10 production. This study seems too incomplete without explanation of IL-10 producing cells after the CD25 depletion. Author should add one more flowcytometry experiment to know about IL10 producing cells after 96 hours culture. This may be helpful to further strengthen the current manuscript.

3. Please also check data analysis test. You should try Wilcoxon T test (Two tailed) for within the groups. 

4. Figures are poorly represented and need major overhauling with respect to style, labelling and font size etc.

5. Graphs need to be more informative like by showing significant differences between the groups. What letter represents what is not mentioned? Before drug (MDT/Prednisolone) and after taking drug graphs should be shown for more clear information. In the present condition, data presentation is very complicated, it needs to be clear by graphical representation.

Reviewer #3: -The objectives of the study clearly articulated with a clear testable hypothesis stated

-The study design appropriate to address objective

-The population clearly described and appropriate for the hypothesis being tested(30 non-reactional LL and 30 LL participants with ENL were enrolled)

- Used correct statistical analysis 

-Properly address ethical issue

**Results**

-Does the analysis presented match the analysis plan?

-Are the results clearly and completely presented?

-Are the figures (Tables, Images) of sufficient quality for clarity?

Reviewer #1: No

Reviewer #2: 1. Manuscript entitled "Regulatory T cells in Erythema nodosum leprosum maintain anti-inflammatory function" was written with the aim to evaluate the suppressive function of Treg cells in LL and ENL. Authors reported that the function of Treg cells remains intact despite their low number in ENL cases. This needs to be explained in the results and discussion accordingly.

2. Authors have evaluated it by depletion of CD25 cells and reported that levels of TNF and IFN are increased in PBMCs of LL cases when stimulated with WCL after MDT treatment. However, it was not significantly different when CD25 was added. 

3. Regarding IL10 level, line no. 253 - 258 needs to be rewritten as there is no difference in their meaning.

Reviewer #3: Results

The analysis presented match the analysis plan and the results clearly and completely presented

-They implement to use figures (Tables, Images) of sufficient quality for clarity 

But in Table 1: Demographic characteristics of study participants LL(n=30)column Sex row Male=27 and Female=7 the sum of sex=34 not 30 

I think it is better to describe figures and tables in the form of self explanatory (What, Where and when..

**Conclusions**

-Are the conclusions supported by the data presented?

-Are the limitations of analysis clearly described?

-Do the authors discuss how these data can be helpful to advance our understanding of the topic under study?

-Is public health relevance addressed?

Reviewer #1: No

Reviewer #2: Need to redone in context to new experiments and inputs suggested accordingly

Reviewer #3: Conclusions

-The paper under the conclusion part addressed by the data presented in well manner 

-The limitations of the study is not clearly described

-The authors discuss how these data can be helpful to and showed the public health relevance in different ways

**Editorial and Data Presentation Modifications?**

Reviewer #1: NA

Reviewer #2: Need Major Revision

Reviewer #3: -Table 1: Demographic characteristics of study participants LL(n=30)column Sex row Male=27 and Female=7 the sum of sex=34 not 30 

-I think it is better to describe figures and tables in the form of self explanatory (What, Where and when.

-Authors used many outdated references

**Summary and General Comments**

Reviewer #1: Negera et al reported an investigational study on LL and ENL PBMC by stimulated cells in vitro. Overall, this study is interesting and showed the relation between CD25+/- cells and cytokines. Several comments should be addressed.

1. The author defined CD25+ PBMC as Treg Cells. However, it is now commonly to use CD4+CD25+foxp3+ to define Treg Cells. Although there may have some publications to support, the author should conduct experiment to show the consistence between two methods.

2. Current results showed relations of selected cytokine expression with stimulated cells from different stages solely from PBMC. As M.leprae mainly affected skin, the immune response in skin was much more important to cover the title “Regulatory T cells in Erythema nodosum leprosum maintain anti-inflammatory function”. Better to add results from skin biopsy.

3. There are two leprosy single cell sequencing analysis have been published in Nat Immunology and Cell discovery. Many aspects of leprosy mechanisms especially immunity have been covered. The author should discuss the current results and published results.

Reviewer #2: 1. Figures are poorly represented and need major overhauling with respect to style, labelling and font size etc.

2. Please add figure of flow cytometry dot plot showing CD25+ and CD25neg depletion assay.

3. Treg cells are major source of IL-10 production. This study seems too incomplete without explanation of IL-10 producing cells after the CD25 depletion. Author should add one more flowcytometry experiment to know about IL10 producing cells after 96 hours culture. This may be helpful to further strengthen the current manuscript.

4. Please also check data analysis test. You should try Wilcoxon T test (Two tailed) for within the groups. 

5. Manuscript entitled "Regulatory T cells in Erythema nodosum leprosum maintain anti-inflammatory function" was written with the aim to evaluate the suppressive function of Treg cells in LL and ENL. Authors reported that the function of Treg cells remains intact despite their low number in ENL cases. This needs to be explained in the results and discussion accordingly.

6. Authors have evaluated it by depletion of CD25 cells and reported that levels of TNF and IFN are increased in PBMCs of LL cases when stimulated with WCL after MDT treatment. However, it was not significantly different when CD25 was added. 

7. Regarding IL10 level, line no. 253 - 258 needs to be rewritten as there is no difference in their meaning. 

8. Graphs need to be more informative like by showing significant differences between the groups. What letter represents what is not mentioned? Before drug (MDT/Prednisolone) and after taking drug graphs should be shown for more clear information. In the present condition, data presentation is very complicated, it needs to be clear by graphical representation.

Reviewer #3: The Title , methodology, significances and the way of writing it was based on scientific manner with advance.

They were addressed all important criteria's of scientific paper steps except limitation 

-Authors used many outdated references ;so it is better use updated references

PLOS authors have the option to publish the peer review history of their article (what does this mean?). If published, this will include your full peer review and any attached files.

Reviewer #1: No

Reviewer #2: Yes: Dr. Rupesh K. Srivastava

Reviewer #3: No
---

## [Decision Letter · Decision Letter 1]

8 Jul 2022

Dear Dr Negera,

We are pleased to inform you that your manuscript 'Regulatory T cells in Erythema nodosum leprosum maintain anti-inflammatory function' has been provisionally accepted for publication in PLOS Neglected Tropical Diseases.

Best regards,

Furen Zhang

Associate Editor

Gerson Penna

Deputy Editor

NA

Reviewer's Responses to Questions

**Key Review Criteria Required for Acceptance?**

**Methods**

-Are the objectives of the study clearly articulated with a clear testable hypothesis stated?

-Is the study design appropriate to address the stated objectives?

-Is the population clearly described and appropriate for the hypothesis being tested?

-Is the sample size sufficient to ensure adequate power to address the hypothesis being tested?

-Were correct statistical analysis used to support conclusions?

-Are there concerns about ethical or regulatory requirements being met?

Reviewer #1: Yes

Reviewer #2: The authors had addressed all of my concerns well.

**Results**

-Does the analysis presented match the analysis plan?

-Are the results clearly and completely presented?

-Are the figures (Tables, Images) of sufficient quality for clarity?

Reviewer #1: Yes

Reviewer #2: The authors had addressed all of my concerns well.

**Conclusions**

-Are the conclusions supported by the data presented?

-Are the limitations of analysis clearly described?

-Do the authors discuss how these data can be helpful to advance our understanding of the topic under study?

-Is public health relevance addressed?

Reviewer #1: Yes

Reviewer #2: The authors had addressed all of my concerns well.

**Editorial and Data Presentation Modifications?**

Reviewer #1: NA

Reviewer #2: Accept

**Summary and General Comments**

Reviewer #1: My comments have been addressed although lack of additional experiments.

Reviewer #2: The authors had addressed all of my concerns well.

PLOS authors have the option to publish the peer review history of their article (what does this mean?). If published, this will include your full peer review and any attached files.

Reviewer #1: No

Reviewer #2: **Yes: **Dr. Rupesh K. Srivastava

---

## [Editor Report · Acceptance letter]

18 Jul 2022

Dear DR Negera,

We are delighted to inform you that your manuscript, "Regulatory T cells in Erythema nodosum leprosum maintain anti-inflammatory function," has been formally accepted for publication in PLOS Neglected Tropical Diseases.

Best regards,

Shaden Kamhawi

co-Editor-in-Chief

Paul Brindley

co-Editor-in-Chief
